# Environmental Protection Is Not Relevant in the Perceived Quality of Life of Low-Income Housing Residents: A PLS-SEM Approach in the Brazilian Amazon

**Wylliam Bessa Santana \* and Luiz Maurício Furtado Maués** 

Programa de Pós-Graduação em Engenharia Civil, Instituto de Tecnologia, Campus Universitário Guamá, Universidade Federal do Pará (UFPA), Belém 66075-110, PA, Brazil
**\*** Correspondence: wylliam.santana@ifpa.edu.br

**Abstract:** Meeting the needs of users is imperative in construction, especially those of low-income people. This research looks into the perceptions of low-income users concerning green building (GB) and discusses how building sustainability can contribute to improving their lives. To this end, a model was developed using partial least squares structural equation modeling (PLS-SEM) relating the perceptions on residents' quality of life with the GB criteria of Blue House Label (Selo Casa Azul—SCA), a Brazilian Sustainable Label. This model was based on data from a survey with 658 residents of the 'Minha Casa, Minha Vida—MCMV' (My Home, My Life) program, which is part of the Brazilian social housing system. The results of the model suggest that intangible issues such as the environmental protection criteria related to the construction of the building are not capable of influencing their perception of quality of life in the project. On the other hand, GB criteria capable of providing more practical benefits to low-income residents were broadly accepted, such as urban quality, GBT related to cost reduction, water management, and social practices. Furthermore, this article contributes to the discussion about sustainable social housing, the importance of adopting social criteria in GB, and the potential of environmental education to contribute to meeting sustainable development goals (SDG).

**Keywords:** low-income housing; social housing; sustainable social housing; green building; sustainable development; sustainable development goals

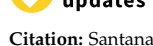



## 1. Introduction

As a popular saying goes, "the customer is always right". This may be true or, at least, meeting the customer's needs may be an effective means of being successful in adopting a product [1]. However, users' needs stem from a variety of factors, and often a lack of believed information can result in misinterpretations and difficulty in accepting a product, as has been reported in relation to green buildings [2–4]. In this regard, studying users' perceptions concerning building sustainability is key to developing contemporary sustainable housing that can overcome the current limitations of sustainable buildings [5]. This concern is more relevant in relation to low-income users, as they are a group that is subject to irregular housing conditions that pose risks to their health and safety and are located on the outskirts of cities without access to education, employment, and income [6–9]. To better understand this scenario and to help in the design of more sustainable construction, this research aims to explore the perceptions of low-income users on GB criteria.

Broadly speaking, the difficulty for low-income populations to access quality housing is a global concern, both in developed and developing countries [10–12]. To overcome this problem, the governments of several countries have created social housing programs [12]. The social gains from these programs are enormous, and the beneficiary population is growing all over the world [6,13–16], as well as the criticism about how these projects are developed, excluding the popular participation in the identification of their needs and

favoring the economy of scale to the detriment of the location of the houses, thus resulting in the isolation of these communities and in the difficulty of generating employment, income, and access to the basic infrastructure of the cities [6,7,17]. Rethinking the current model of social housing by incorporating the users' needs and values is key to bringing real benefits to the quality of life of this population [6].

More than this, because of the scale of these programs and the beneficiary population, there is a growing understanding in the scientific literature that rendering social housing more sustainable is the only way to achieve sustainable development in cities [13,15]. This movement gained even more strength with the 2030 Agenda, a document signed by the leaders of 197 countries that defined 17 sustainable development goals (SDGs) and targets to be reached from 2016 to 2030 [18]. For the United Nations, the SDGs "are a model for achieving a better and more sustainable future for everyone" and address global challenges such as poverty, inequality, climate change, environmental degradation, peace, and justice [19].

The SDGs have brought urgency to the discussion about building sustainability, as they set targets to be met by 2030. In this regard, not only the need to facilitate access to quality housing but also to facilitate access to sustainable housing is currently discussed [14–16], thus contributing to obtaining benefits for the users. Sustainable social housing can help reduce human impacts on nature, reduce social inequalities, and provide financial savings for low-income residents while helping pay for their own housing [13,15,20].

However, there is no simple way to incorporate sustainability aspects into social housing, as it must reconcile the high initial cost of these initiatives with the need to provide low-cost housing [16,21,22]. Solving this issue can be challenging, and part of the solution requires choosing green building technologies (GBT) that meet the economic, social, and environmental needs of low-income communities [15].

In view of the need for sustainable social housing, this research aims to identify the most relevant criteria of green building for low-income communities, thus allowing for identifying those strategies that foster more sustainable communities and for providing important insight into the importance of environmental protection, as perceived by the low-income population.

## 2. Literature Review

### 2.1. The Importance of User's Opinion for Sustainable Social Housing

The user plays a decisive role in the sustainability of social housing. After all, the user is responsible for the operation of the building, the phase in which most of the building's energy consumption and greenhouse gas emissions occur [23], especially in green buildings, where operations are based on technologies that involve complex procedures and require behavioral changes on the part of the users to provide the expected efficiency gains [24].

Using new technologies in social housing may then conflict with users' habits and interests, thus contributing to the creation of a gap between design and real performance [25,26]. The most notorious example of this gap is the so-called "rebound effect", whereby, for example, the purchase of an energy-efficient technology results in increased energy consumption by the user [27]. Moreover, in the last case, the lack of information and training on the part of the users to manage their household can even result in the abandonment or replacement of the technology by other less efficient ones but of known use by the dweller [28].

One of the measures to promote the efficiency of green building technologies in social housing is to improve the users' behavior in using these technologies. This is because green technologies are significantly more complex and require more knowledge and engagement by the dweller than conventional building technology [25]. As a consequence, the low-income user is not always prepared to deal with these technologies, and adopting them can promote in the dweller a loss of identity regarding their household and a lack of efficiency in the system [24]. Public policies of environmental regulation, information, and education of the population are some strategies to encourage user behavior toward the efficiency

of green technology in housing; however, the results thereof are still limited and are not necessarily translated into behavioral changes and benefits by using that technology [29].

Another measure to promote the efficiency of green building technologies is to promote the acceptance of these technologies, for example, by investigating the users' opinions on what green technologies are most suitable for their households. This measure can support more functional and contextualized choices about the technologies being employed, thus assisting in the development of social housing-oriented strategies that provide the greatest benefits for sustainable development with the least amount of economic and technological resources [26,30]. For this, it is necessary to investigate what green technologies are best adapted to the users' needs and to incorporate them into the housing design phase in order to render the best benefits for sustainable development compatible with the economic standards of social housing.

### 2.2. Low-Income Communities

The rapid industrialization over the past decades, with the growing need for labor to supply the industries in the cities and the automation of the countryside, resulted in a mass migration of the population from the countryside to the cities seeking better living conditions. However, the pressure of this phenomenon on urban density resulted in the segregation of the poorest population into informal housing on the outskirts of the cities [31]. Despite the efforts of the public sector to promote access to housing for the low-income population in the cities, the inability to solve this problem has led to the proliferation of slums around the world, especially in the global south [12]. In these slums, informal housing and inadequate basic infrastructure generate dangerous environments and social inequalities and shrink the income of dwellers, thus hindering access to health, education, and employment [7–9,32].

Remarkably, despite the daily difficulties that the population of these slums experience, a commonplace fact that makes the headlines in these countries is the capability of community liaison for the common welfare, a fact that takes place even during the current pandemic period, when, for example, residents unite to buy and produce food for the needy [33,34]. This can be observed in other situations and dates back to the origins of most slums when the residents come together through a great collective effort to occupy a space and build their households [15,35]. This capability of the low-income population to organize themselves as a community to address a specific issue has been gaining focus in the scientific literature through the view that it can contribute to solving the macro-problems of cities [36,37], including transformation towards sustainable development.

According to Sullivan and Ward [15], these communities have a great capability for collective effort, which may lead them to become more sustainable, provided that they have the conditions to do so by themselves, such as having access to low-cost GBT. As for concrete facts, Reyes [38] describes the importance of popular participation for the success of Mexico's sustainable social housing, as well as its weakening as the community movement has become mischaracterized and housing organizations have lost autonomy.

In addition to the previously discussed contributions, this research also contributes to the discussion about the ability of low-income population communities to organize themselves to meet the SDGs and achieve a more sustainable future by 2030.

### 2.3. Blue House Label (Selo Casa Azul—SCA)

GB labels are structured models for construction projects to meet sustainability aspects [39,40]. They are adopted as a way to confer commercial advantages to buildings that adopt sustainable practices [41]. These labels can be either international or domestic, depending on their acceptance (international labels certify buildings in several countries, while domestic labels certify buildings only in one country or region of similar culture (The ESTIDAMA GB label, for example, has certified buildings in more than one country. However, it is not usually highlighted in the literature as an international label because it has limited acceptance and certifies buildings only in Arab countries of similar cul-

ture [42])) and the existence of certifying agencies in the countries [43]. As an example of international seals, the most known ones are the Leadership in Energy and Environmental Design (LEED) and the British Building Research Establishment Environmental Assessment Method (BREEAM).

These international labels have an important pioneering role and have introduced GB to most developing countries, which in time, tend to formulate their own labels [43]. Domestic labels tend to be developed in order to make the GB international model compatible with the needs of each country. In Brazil, the SCA was created in 2010 by Caixa Econômica Federal, a government institution that has the largest housing investment portfolio in the country. The SCA was made aiming to reduce the environmental impact of the projects financed by the bank by encouraging the implementation of green techniques and materials that mitigate the impacts caused both during and after the completion of the construction works.

According to Caixa, the SCA is a GB label created entirely for the Brazilian reality. However, several of its criteria are similar to those adopted by international labels, such as the LEED [44]. The main difference lies in the criteria related to social practices because in the Brazilian label, there are eleven criteria related to social practices, while in LEED for Homes v4, there is only one, and in BREEAM International new constructions, there are none. This fact contributes to SCA as the best option for housing in the country [44,45], especially considering the Brazilian reality [46].

Currently, the SCA has a low number of certifications in the country, far behind LEED, for example. Up until June/2021, the SCA certified 60 projects [47], while LEED has already certified 681 projects in Brazil, 99 of which in 2020 alone [48]. When aiming to increase the number of certifications, the Brazilian label published a new successful version, called Blue House Label+ (Selo Casa Azul+), in July/2020, and after just under a year, it was responsible for the certification of 43 projects. To this end, the update made its means of certification more flexible, which in addition to certifying projects that meet an overall minimum score, included the #plus certification when the project achieves a score in two or more categories, in addition to the mandatory criteria. Furthermore, the new update now requires compliance with the recent Brazilian building performance standard NBR 15.575 [49] and includes a criteria category called "innovation", following the example of international labels such as LEED and BREEAM.

Because it is a label developed exclusively for the Brazilian scenario and because of its good coverage of criteria related to the social axis of sustainability, this research adopted the SCA criteria as a sustainability indicator to be used to identify the perception of social housing residents in Brazil. This was adopted based on a more realistic view of social housing in Brazil, inasmuch as meeting this label would be closer to what is possible under the MCMV's budget limitations, to the detriment of more ambitious sustainability criteria such as ecology, health, or well-being. In addition, the 2010 SCA has been adopted, and not the Blue House Label+, since this research commenced in 2019, thus prior to the "plus" version of the Brazilian label.

## 3. Method

To meet the research objective, that is, to identify the most relevant green building criteria for low-income communities, quantitative exploratory research techniques were used based on survey and structural model evaluation (SEM) techniques. The following topics describe the steps of the method that was adopted.

### 3.1. Survey

A questionnaire-based survey was conducted to obtain the data used in the research herein. The questionnaire had three distinct fields, namely: the first contains a brief characterization of the interviewee by age and gender; the second is a single question about the quality of life in the housing development as perceived by the resident, structured on a 5-level Likert scale ranging from "very bad" to "very good"; the third is intended to

identifying the perception of the residents about the importance of the GB-related criteria required by the SCA, and consists of 21 closed, scaled questions (5-level Likert scale) ranging from "unimportant" to "very important". This scale-based model was adopted in order to obtain the best results by means of the recommendation-based methodology being used [50,51] and research with similar methodologies [52,53]. Table 1 shows the questions that were used.

**Table 1.** Constructs, indicators and questions adopted.

| Construct | Indicators | | |
| --- | --- | --- | --- |
| | Acronym | Question | Benefits |
| 6—Perception of the quality of life in the housing development | PQLH1 | How do you rate your quality of life in relation to this housing development? | |
| Introductory question to all other questions | | Consider whether the correct application of these items, even if not currently in place, would be important in promoting a better sense of well-being for you in your current household | |
| 1—Facade maintenance-related savings | PFMRS1 | The facade of the house with cladding that will last at least 15 years | Reducing the use of non-renewable materials, waste generation, and costs resulting from frequent facade maintenance |
| 2—Savings for the resident | PSR1 | Measures in place to improve energy savings, such as installing energy-saving light bulbs in the households and sensors in shared areas | Reducing electrical energy consumption and costs by using efficient light bulbs and appliances |
| | PSR2 | Distribution of appliances with the A * level efficiency seal, which are more cost-effective, to the residents in the housing development | |
| | PSR3 | An alternative power generation system, such as solar energy, in place to supply part of the energy consumed in the households | Cost reduction for the resident and cleaner and more sustainable energy generation |
| 3—Sustainable leisure equipment in the housing development | PSLEH1 | Bike racks in place and bike lanes in the housing development | Encouraging the use of healthier and more environment-friendly transportation |
| | PSLEH2 | Equipment or spaces, such as woods, sports court, gym, game room, playground etc. in place in the housing development compatible with the amount of households | Encouraging healthy practices of coexistence and entertainment for the residents |
| 4—Water management | PWM1 | Systems in place that help save water, such as a 3- and 6 L flush system for the toilets and water-saving devices for the faucets and showers | Reduction in water and natural resource consumption |
| | PWM2 | A system in place to harness rainwater for secondary use | |
| | PWM3 | Permeable areas on the ground so that rainwater can seep and prevent flooding | Preventing the risk of floods and reducing the overload of public drainage networks. |
| 5—Piped gas and water heating | PPGWH1 | Water heating system in place | Reducing gas and/or electricity consumption |
| | PPGWH2 | Piped gas in place in the household | Reducing gas consumption |
| 7—Environmental protection | PEP1 | A specific space in place for selective garbage collection in the housing development | Promoting reuse of non-renewable materials |
| | PEP2 | Living in a house built with a view to environmental care, such as using certified materials, reduced waste generation, recycling and reuse, and the correct disposal of debris | Reducing consumption of non-renewable materials; reducing construction waste disposal; increasing building durability and promoting environmental awareness |
| 8—Social practices | PSP1 | Relying on the participation of future residents in planning for the design of the houses and of the housing development | Stimulating the permanence of the residents and the valuation of the property |
| | PSP2 | Having activities in place on environmental education and sustainability for the residents addressing, for example, selective waste collection, rational use of water, energy saving etc. | Promoting environmental awareness among residents |
| | PSP3 | Residents being trained in the management of the project, so that they can play a more active role in the neighborhood associations | Greater inclusion of the local community in project management and decision-making |
| | PSP4 | Activities in place for personal and professional development of the residents, such as literacy classes, digital inclusion, cultural activities, and vocational courses | Generating jobs and income in the community; Fostering the health and well-being of the residents |
| 9—Urban quality | PUQ1 | The surroundings of the housing development should have a good basic infrastructure, regular public transportation, businesses such as a drugstores, markets, schools, hospitals etc | Facilitating access to city infrastructure; Promoting employment, income, health, welfare, and education for residents |

* seal awarded by the Brazilian Energy Efficiency Information Center for highly energy-efficient household appliances [54].

The application method adopted was face-to-face interviews since this method secures greater reliability of results and solves the problem of the low percentage of responses from self-evaluation surveys in the region [55].

For validation of the questionnaire, the pre-test method was adopted, followed by an interview, whereby a preliminary questionnaire was administered to people representing the population under study in as many rounds as possible [56,57]. This validation was performed in 3 rounds of interviews with residents of one of the households under analysis; in the first round, 1 person was interviewed and corrections were made; in the second round, 3 residents were interviewed, and corrections were made; and in the third round, 5 residents were interviewed. Since no new deviations in understanding were identified, the authors considered the validation sufficient.

### 3.2. Data Collection

Data collection was performed through post-occupancy interviews. This format was chosen to take advantage of the experience of the residents in the projects, as it is a well-experienced method in sustainability studies of housing projects [58–60]. As well as for reducing the bias that could arise from interviewing other sample groups such as workers or stakeholders, especially considering that the sample is made up of low-income people with limited access to education and for whom housing has a major impact on their quality of life [9,32].

The data were collected by two groups of students, one group consisting of five graduate students in civil engineering and the other being a group of thirteen technical level students in building construction enrolled in a research project designed for this purpose. In order to prepare these students, meetings were held with the professors (the authors of the research herein) in order to make them understand the research objectives and the theoretical concepts being addressed. The data were obtained in five cities in the state of Pará, north of Brazil, in the middle of the Amazon region. These cities include the state capital and four other cities, all of which are ranked among the state's seven most populated cities, which together account for 31% of the state's population [61].

The minimum sample size was 371 interviews, as defined according to Ayres, Furlaneto, and Ayres [62]; considering a 5% margin of error and a population of 11,233 inhabitants benefited in the state by the MCMV, the Brazilian government's social housing program. Several visits were made to the housing projects from July to October 2019, the first ones being supervised by the authors herein. Finally, 658 complete replies were obtained.

### 3.3. Data Analysis

In order to explore the perception of the social housing user, a model was developed by using partial least squares structural equation modeling (PLS-SEM) by relating the perceptions of the SCA sustainability criteria with perceptions of quality of life in the household. PLS-SEM was adopted because of its broad acceptance in exploratory research, even when the data are non-normally distributed, and the conceptual model is complex [50], as is the case in this research. Structural equation modeling (SEM) is capable of conducting both a confirmatory factor analysis (CFA) and a path analysis in a single structural equation model [63,64]. PLS-SEM is a multivariate analysis technique that allows not only for the combination of variables into factors but mainly tests theoretical hypotheses about the relationships amongst the resulting factors [50]. To this end, many papers have successfully adopted PLS-SEM involving exploratory research on user perceptions [52,53,63,64].

In PLS-SEM, on-site measured indicators make up constructs, and hypothetical relationships between these constructs are tested [50]. As for their formation, constructs can be either formative or reflective. In the herein research, only reflective constructs are adopted, as it is the constructs that cause the items [50,64]. The following topic introduces the development of the constructs and the research hypotheses.

### 3.4. Development of Hypotheses

Those criteria that are relevant to the Brazilian socio-environmental label SCA were used herein as dimensions of the sustainability of social projects. The SCA was chosen because it is a Brazilian green building label adapted to national characteristics [45,65], which can be employed without the need for large investments [46] and because it envisions social aspects [66], and, according to our viewpoint, can be a great differential when applied in social housing. Furthermore, the SCA was developed by Caixa Econômica Federal, the same public institution responsible for contracting the MCMV works, thus strengthening the possible compatibility with the program.

Based on the SCA sustainability criteria, the indicators were formulated to harness users' perceptions. A rigorous procedure was then developed for making the constructs that define the research hypotheses. After collecting the data, the indicators were grouped according to the following steps:

1. A factor analysis of main components was performed in order to group the indicators into a smaller number of components. This step observed the procedure of Ping et al. [67]. However, it can be challenging to interpret some of the components obtained via factor analysis [68], and, in the case of this research, some components, apart from forming incoherent groups of indicators, did not meet the CFA criteria required by PLS-SEM [50];
2. In order to correct this inconsistency, some components were modified, whereas others were created based on a content analysis [69,70];
3. CFA is a critical step of PLS-SEM [61] and was performed in order to confirm whether the components in steps (a) and (b) meet the PLS-SEM criteria [50], and if not, step (b) was repeated, followed by a new CFA until a set of components, consistent with the research objective, was obtained.

The final result of preparing the constructs is shown in Table 1. Based on these, it was possible to formulate the following research hypotheses:

**H1.** *The savings related to facade maintenance positively affect the perceived quality of life in the housing development;*

**H2.** *The savings for the resident positively affect the perceived quality of life in the housing development;*

**H3.** *Sustainable leisure equipment in the housing development positively affects the perceived quality of life in the housing development;*

**H4.** *Water management positively affects the perceived quality of life in the housing development;*

**H5.** *Piped gas and water heating positively affect the perceived quality of life in the housing development;*

**H6.** *Environmental protection positively affects the perceived quality of life in the housing development;*

**H7.** *Social practices positively affect the perceived quality of life in the housing development;*

**H8.** *Urban quality positively affects the perceived quality of life in the housing development.*

### 3.5. Conceptual Model

Finally, once the research hypotheses were defined, the conceptual model presented in Figure 1 could be drafted. Here, one can see the constructs of the model developed in topic 3.4 (above), the indicators that compose the constructs (for more details, see Table 1), and the research hypotheses.

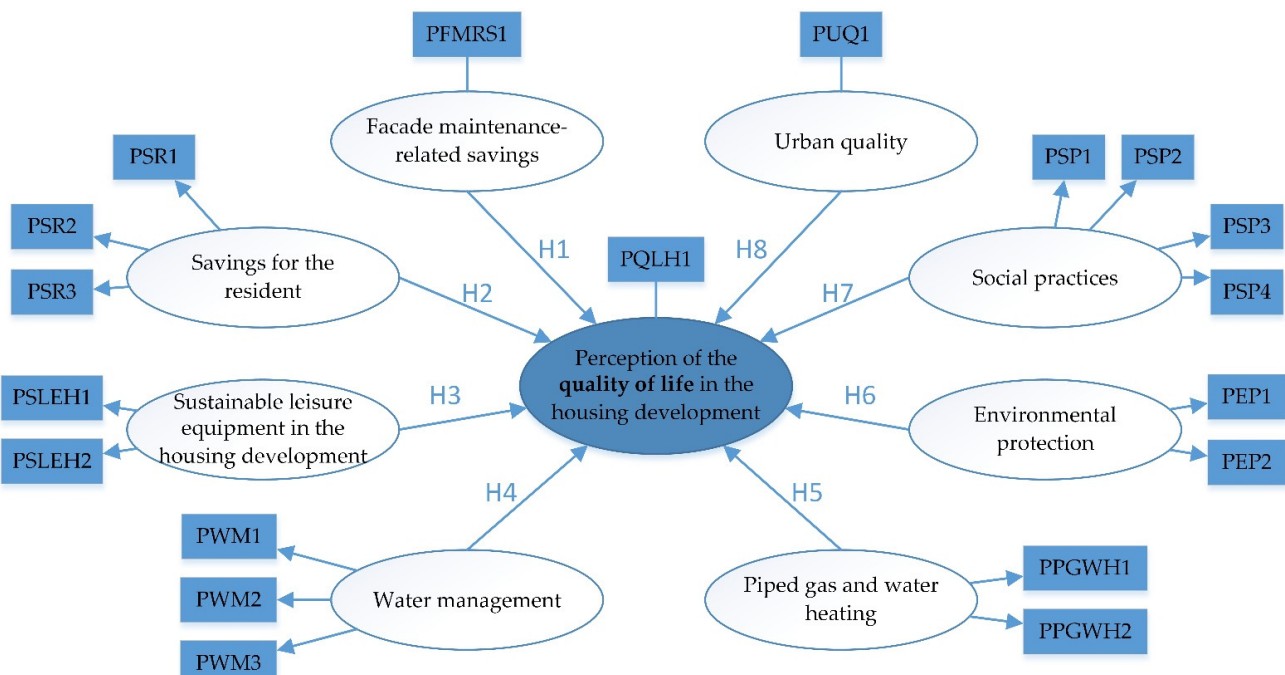

**Figure 1.** Conceptual model of the study.

According to the research objective, it can be observed that the hypotheses are aimed at investigating the relationship between low-income users' perception of GB criteria (constructors) and their perception of quality of life in housing development. As for the relationships between constructs and indicators, arrows are observed coming out of the constructs toward the indicators. This represents the reflexive relationships between these elements, existing when it is the constructs that cause the items, as discussed in topic 3.3.

However, there are no arrows connecting the indicators and the "perception of quality if in the housing development", "façade maintenance-related savings" and "urban quality" constructs. As there is only one indicator, it is not possible to evaluate the relationship between the indicators; thus, there is no reason to differentiate whether there is a formative or reflective relationship between them.

The results of the model and hypothesis testing are presented in the following section.

## 4. Results

Based on the proposed method, this chapter introduces the sample breakdown, the measurement model evaluation—the variance-based statistical analysis that assesses the quality of the PLS-SEM model—and, finally, introduces the structural model evaluation with the results of the bootstrapping procedure.

### 4.1. Sample Details

This research obtained a total of 658 questionnaires applied in thirteen sets of MCMV beneficiaries, a number that is considerably larger than the required sample size of 371 questionnaires. The residents who were interviewed were 34 years old on average and have lived in the housing developments for an average of 3.8 years. As for gender, 57% of the respondents were male, and 43% were female.

Table 2 shows some characteristics of the housing developments that were evaluated. It shows that the research sought good sample representativeness, obtaining data from 12 housing developments located in several cities in the state of Pará, northern Brazil. The households vary between houses and apartment buildings, with delivery dates from the second year of the PMCMV in the state (2011) to the year the data were collected (2019). As for the average time of occupancy per household, values ranging from 1 month to 6.7 years



can be observed, thus corroborating the variability of the sample group since this study sought to investigate the perception of residents with varied times of housing occupancy.

**Table 2.** Details of the sample per housing development.

| Development | Household Type | No. of Housing Units | Built-Up Area (m$^2$) | Year of Delivery | Average Occupancy (Years) |
|---|---|---|---|---|---|
| 1 | Houses | 456 | 33.69 | 2013 | 4.3 |
| 2 | Apartment Block | 500 | 44.6 | 2011 | 3.9 |
| 3 | Houses | 102 | 36.4 | 2013 | 4.0 |
| 4 | Apartment Block | 496 | 41.16 | 2013 | 3.4 |
| 5 | Houses | 222 | 36.4 | 2014 | 2.5 |
| 6 | Apartment Block | 2720 | 39 | 2019 | 0.1 |
| 7 | Apartment Block | 384 | 39.74 | 2019 | 0.7 |
| 8 | Houses | 1000 | 39 | 2015 | 3.3 |
| 9 | Houses | 499 | 41 | 2011 | 5.7 |
| 10 | Apartment Block | 332 | 33.69 | 2013 | 6.7 |
| 11 | Houses | 50 | 41 | 2019 | 0.4 |
| 12 | Houses | 1090 | 33.69 | 2012 | 5.7 |

*4.2. Evaluating the Measurement Model*

This is an important SEM step developed to confirm and refine the measurement model, i.e., the items (components) and the constructs (latent variables) [63]. Evaluating the measurement model aims to verify the suitability of the measurement model for path analysis [52], as it helps ensure that the constructs, the relationships of which being the basis of the model, are being correctly represented and measured.

This step is formed by the evaluation of four criteria, namely: internal consistency, convergence validity, discriminant validity, and indicator reliability [50]. CFA was conducted in this study through the SmartPLS (v. 3.3.3) software. CFA is a broadly used technique in the SEM and aims to test whether the arrangement of indicators into constructs is consistent with a factor model [51]. The results are shown in Tables 3 and 4 below.

Internal consistency is usually given by composite reliability; this value ranges from 0 to 1, and the closer to 1, the higher the reliability. What is desired, however, are values between 0.7 and 0.9 since very high values can mean redundancy of the measurement items [50]. In this sense, the results shown in Table 3 denote good internal consistency, with values ranging from 0.702 to 0.849.

It is important to note that constructs with only one indicator (constructs 1, 6, and 9) are not evaluated as to the measurement model since there are no other indicators to relate to.

The convergence validity is evaluated through the average variance extracted (AVE), and the main rule of thumb is that the AVE should be greater than 0.5, thus indicating that, on average, the constructs explain more than half of the variance of the indicators [51]. Table 3 shows the AVE results for the constructs, ranging from 0.502 to 0.694, thus confirming the model's convergence validity.

The discriminant validity is assessed via two methods, namely: the Fornell–Larcker criterion is the more conservative one, and the main rule of thumb is that the square root of each construct's AVE must be larger than its correlation with other constructs [50]. The Fornell–Lacker matrix for this research is shown in Table 3, where you can see that there is good discriminant validity since the square roots of the variance (highlighted diagonal) are much higher than the correlations between constructs.

Another method of checking discriminant validity is by examining the cross-loadings. In this case, all of the indicators must have greater loadings with the associated construct [50]. Table 4 shows the cross-loadings between indicators and constructs. Furthermore, it can be seen that the rule for discriminant validity has been satisfied.

**Table 3.** Results of measurement model evaluation.

| | 1 | 2 | 3 | 4 | 5 | 6 | 7 | 8 | 9 |
|---|---|---|---|---|---|---|---|---|---|
| 1—Facade maintenance-related savings | 1 | | | | | | | | |
| 2—Savings for the resident | 0.224 | 0.724 | | | | | | | |
| 3—Sustainable leisure equipment in the condominium | 0.098 | 0.153 | 0.756 | | | | | | |
| 4—Water management | 0.137 | 0.328 | 0.191 | 0.709 | | | | | |
| 5—Piped gas and water heating | 0.036 | 0.212 | 0.121 | 0.137 | 0.833 | | | | |
| 6—Perception of the quality of life in the housing development | 0.028 | 0.043 | −0.149 | −0.182 | −0.043 | 1 | | | |
| 7—Environmental protection | 0.212 | 0.392 | 0.201 | 0.42 | 0.202 | −0.055 | 0.754 | | |
| 8—Social practices | 0.321 | 0.4 | 0.238 | 0.391 | 0.26 | −0.128 | 0.417 | 0.765 | |
| 9—Urban quality | 0.191 | 0.093 | 0.174 | 0.027 | 0.026 | −0.092 | 0.164 | 0.149 | 1 |
| Composite Reliability | 1 | 0.763 | 0.702 | 0.731 | 0.815 | 1 | 0.705 | 0.849 | 1 |
| Average Variance Extracted (AVE) | 1 | 0.524 | 0.571 | 0.502 | 0.694 | 1 | 0.569 | 0.585 | 1 |

Highlighted values on the diagonal are the square root of the AVE.

**Table 4.** Cross loadings between indicators and constructs.

| | 1 | 2 | 3 | 4 | 5 | 6 | 7 | 8 | 9 |
|---|---|---|---|---|---|---|---|---|---|
| PFMRS1 | 1 | 0.224 | 0.098 | 0.137 | 0.036 | 0.028 | 0.212 | 0.321 | 0.191 |
| PSR1 | 0.229 | 0.608 | 0.096 | 0.306 | 0.225 | 0.012 | 0.404 | 0.342 | 0.143 |
| PSR2 | 0.198 | 0.867 | 0.098 | 0.273 | 0.219 | 0.042 | 0.299 | 0.352 | 0.052 |
| PSR3 | 0.102 | 0.671 | 0.153 | 0.188 | 0.045 | 0.027 | 0.25 | 0.216 | 0.067 |
| PSLEH1 | 0.137 | 0.264 | 0.454 | 0.305 | 0.247 | −0.043 | 0.288 | 0.327 | 0.06 |
| PSLEH2 | 0.068 | 0.092 | 0.967 | 0.122 | 0.063 | −0.152 | 0.139 | 0.168 | 0.174 |
| PWM1 | 0.189 | 0.37 | 0.141 | 0.514 | 0.222 | 0.006 | 0.432 | 0.367 | 0.075 |
| PWM2 | 0.126 | 0.348 | 0.162 | 0.504 | 0.101 | −0.024 | 0.313 | 0.321 | 0.038 |
| PWM3 | 0.133 | 0.308 | 0.182 | 0.994 | 0.137 | −0.187 | 0.411 | 0.377 | 0.025 |
| PPGWH1 | 0.024 | 0.179 | 0.105 | 0.133 | 0.969 | −0.046 | 0.185 | 0.235 | 0.028 |
| PPGWH2 | 0.059 | 0.221 | 0.118 | 0.091 | 0.67 | −0.015 | 0.168 | 0.223 | 0.01 |
| PQLH1 | 0.028 | 0.043 | −0.149 | −0.182 | −0.043 | 1 | −0.055 | −0.128 | −0.092 |
| PEP1 | 0.247 | 0.208 | 0.287 | 0.195 | 0.093 | −0.02 | 0.486 | 0.275 | 0.136 |
| PEP2 | 0.15 | 0.365 | 0.123 | 0.402 | 0.193 | −0.055 | 0.949 | 0.37 | 0.135 |
| PSP1 | 0.246 | 0.308 | 0.158 | 0.278 | 0.271 | −0.093 | 0.308 | 0.763 | 0.12 |
| PSP2 | 0.285 | 0.348 | 0.162 | 0.299 | 0.209 | −0.078 | 0.396 | 0.72 | 0.092 |
| PSP3 | 0.317 | 0.342 | 0.145 | 0.305 | 0.258 | −0.074 | 0.321 | 0.796 | 0.119 |
| PSP4 | 0.18 | 0.26 | 0.233 | 0.31 | 0.105 | −0.128 | 0.278 | 0.777 | 0.119 |
| PUQ1 | 0.191 | 0.093 | 0.174 | 0.027 | 0.026 | −0.092 | 0.164 | 0.149 | 1 |

At the indicators level, the indicator reliability and the loading of the indicator with the construct should be higher than 0.55, suggesting that at least 30% of the indicator's variance is related to the construct [71]. However, according to Heir et al. [50], in exploratory research, it is common to keep indicators with low loadings (from 0.4 to 0.7), and it is recommended that exclusion thereof must be made only in cases where the AVE is not satisfactory (≥0.5). In this sense, even though four loadings have values lower than 0.55, all of them were retained because the constructs have AVE higher than 0.5 (Table 3).

*4.3. Evaluating Structural Model*

The research hypotheses are tested in this step. True hypotheses are those with *t*-values 1.96 and 2.57 or greater, at 5% and 1% significance, respectively [50]. Once again, to evaluate the structural model, the software SmartPLS (v. 3.3.3) was used, and the bootstrapping procedure was adopted with a pre-defined number of 5000 samples. The results are shown in Table 5. Out of the eight hypotheses being tested, five were confirmed, which demonstrates that the sustainability criteria H1, H2, H4, H7, and H8 positively influence

the resident's perception of the quality of life in the housing development. Furthermore, from these, criteria H1, H4, and H8 obtained significant path coefficients even at 1%, thus indicating more power of influence than the others. On the other hand, the sustainability criteria H3, H5, and H6 did not obtain significant path coefficients even at 5%, suggesting no influence on residents' perception of the quality of life in the housing development.

**Table 5.** Results of structural model evaluation.

| Hypothesis | Criterion Related to the Perception of the Quality of Life | Path Coefficient | SD | *t*-Value | *p*-Values | Result |
|---|---|---|---|---|---|---|
| H1 | Facade maintenance-related savings | 0.077 | 0.029 | 2.641 | 0.008 | Supported |
| H2 | Savings for the resident | 0.136 | 0.07 | 2.067 | 0.039 | Supported |
| H3 | Sustainable leisure equipment in the housing development. | 0.102 | 0.087 | 1.242 | 0.214 | Rejected |
| H4 | Water management | 0.159 | 0.072 | 2.578 | 0.01 | Supported |
| H5 | Piped gas and water heating | 0.019 | 0.045 | 0.298 | 0.765 | Rejected |
| H6 | Environmental protection | 0.016 | 0.062 | 0.562 | 0.574 | Rejected |
| H7 | Social practices | 0.109 | 0.048 | 2.291 | 0.022 | Supported |
| H8 | Urban quality | 0.075 | 0.029 | 3.007 | 0.003 | Supported |

For ease of understanding, Figure 2 shows a graphic representation of the structural model. It shows the *t*-values obtained for the research hypotheses and the loadings of the indicators with each construct. It is important to note that the greater the loading, the more variance of the construct is explained by the indicator [50]. In this sense, the "environmental protection" (H6) construct, for example, is considerably more explained by the indicator WBS2 than by WBS1, which have loadings of 0.949 and 0.486, respectively. The discussion about this and the other hypothesis test results is presented in the following chapter.

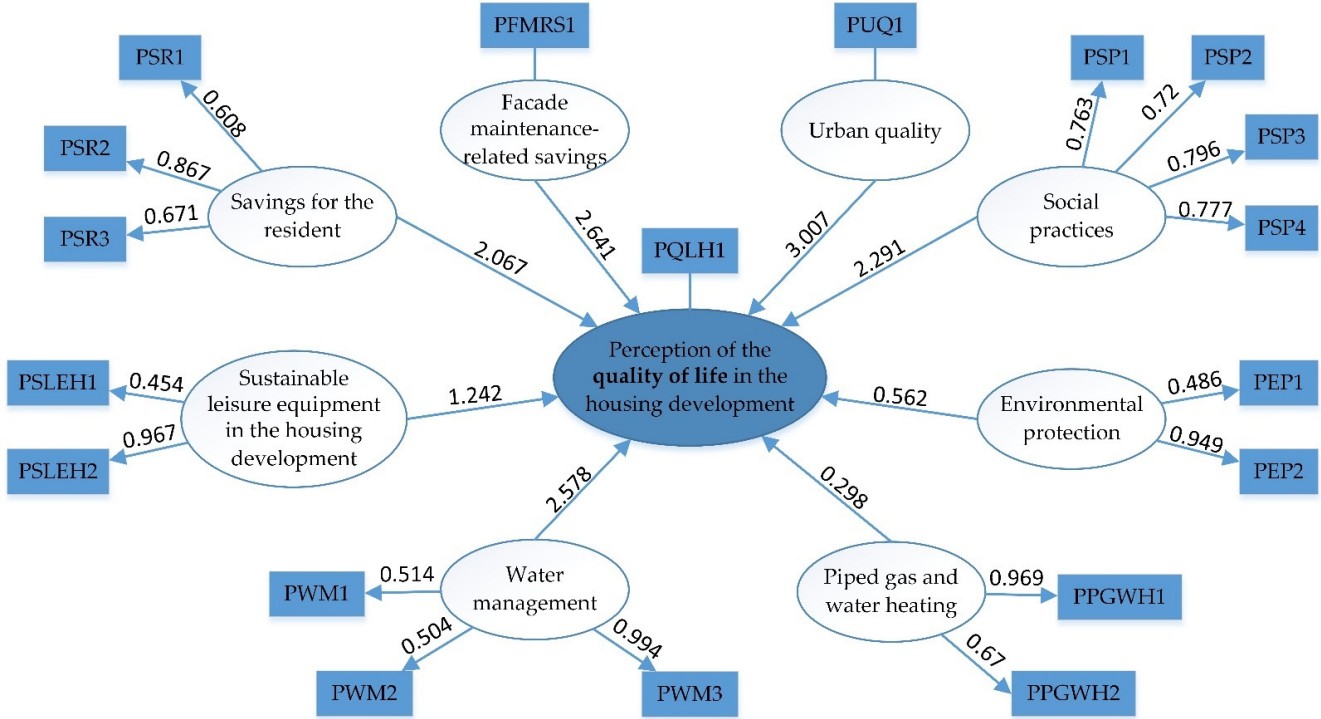

**Figure 2.** Graphic representation of the structural model.

## 5. Discussion

This study contributes to the literature by developing a conceptual model that assesses perceptions of GB criteria by low-income people. As a result, several discussions can be held.

### 5.1. Environmental Protection

Despite the known importance of "environmental protection" for sustainable development [18,19], for GB [2,3,16] or even for the economy [72,73], hypothesis H6 was rejected and stands out with the least influence on residents' perceived quality of life, showing *t*-value of 0.562 and *p*-value of 0.574. This could be worrisome, as it would indicate the residents' disregard for environmental problems. However, for the authors, this result is more related to the nature of the criteria that make it up. It is composed of two criteria, PEP1 (equipment for selective garbage collection in the housing development) and PEP2 (living in a house built with environmental care in mind, such as the use of certified materials, waste reduction, recycling, and reuse, and correct disposal of debris), which have loadings of 0.486 and 0.949, respectively. The higher loading of PEP2 means that this indicator has a greater explanation of the result of the hypothesis [50]. This indicator, in turn, is made up of a set of SCA criteria that indicate GBT related to environmental protection applied during the process of project construction. In this regard, the rejection of the hypothesis can be understood as an inability of the residents to perceive the "environmental protection" related to the construction process and may be associated with a lack of knowledge about the construction process or the difficulty in perceiving sustainability related to aspects that do not translate into clear benefits for the resident. In other words, the model results suggest there is a greater predisposition of these residents to understand environmental protection related to more tangible aspects of sustainability, such as economic, social, water management, and urban quality.

In fact, other research has found similar reactions to environmental issues from low-income housing users. According to Zhang et al. [74], environmental issues are only indirectly related to the intentions of Chinese youth to buy GB, the main driver being the existence of financial incentives, without which most refrain from buying GB. The same was identified in Israel's suburbs, where buyers have little interest in environmental issues and the only goal is to buy low-priced housing [60].

Concerning social housing, Kowaltowski et al. [75] observed a similar relationship; according to the study, the concept of environmental sustainability for these residents is more linked to household savings than to a strong sense of environmental concern. However, even though it is a recurring argument in articles in the area, many of these suggestions are based on assumptions, and little empirical evidence can be found in the literature about the relationship between environmental protection and other sustainability-related aspects. In this regard, this research contributes by clarifying this issue.

### 5.2. Savings for the Resident

As for the economic aspect, hypotheses H1 and H2, "savings related to facade maintenance" and "savings for the resident", were supported by the model, showing *t*-values of 2.641 and 2.067 and *p*-values of 0.008 and 0.039, respectively. This result is in line with the scientific literature [60,74,75], thus demonstrating the predominance of the perception of economic aspects over environmental aspects (hypothesis H7 rejected). This can be easily understood. After all, every economy is important for social housing users, who are often led to join one of these programs by necessity or by the lack of financial conditions to have a house with minimum habitability conditions [6,13,14].

More than that, during the interviews, many residents complained about the increase in household costs in the new households, resulting from the need to pay electricity and water bills and the symbolic portion of the program's benefit, which often was not paid before because they lived in irregular housing or without access to some of these resources (mainly piped water).

In this regard, the results further enhance this discussion, as it corroborates the research hypothesis that GB labels can have good adherence with mass social housing. GB labels can contribute to the household economy of the resident and, therefore, provide favorable conditions not only for a better perception of the quality of life of this resident (according

to the conceptual model shown in Figure 1) but also can help reduce the impacts of social inequalities and encourage the acceptance of these residents to social housing programs.

### 5.3. Social Practices

Similarly, hypothesis H7—"social practices"—was supported, as it has a *t*-value of 2.291 and a *p*-value of 0.022, thus demonstrating that criteria that are more related to social practices positively influence the perception of quality of life of these residents. This was expected since the criteria related to this hypothesis represent clear benefits to the residents, namely: professional training, environmental education, and participation of the residents in project design.

This result is connected to the difficulty of access to services, employment, and income by the residents of social housing, prompted by the isolation and distance from urban centers that these projects are subject to, and result in the abandonment and conversion of housing and public spaces into commercial enterprises in order to satisfy these needs [32,76].

In this regard, the confirmation of this hypothesis suggests that the level of education and qualification of the residents can help reduce the employment and income problem in these communities. As well as the popular participation in the project, thus allowing, for example, the choice of houses and not vertical buildings, the availability of spaces intended for the creation of markets, and the rendering of services by the residents.

### 5.4. Urban Quality

With the highest *t*-value, hypothesis H8—"urban quality"—is the criterion that most positively influences the residents' perception of the quality of life. This fact clarifies the need to provide the basic infrastructure that serves the low-income population. Indeed, providing basic infrastructure for low-income populations is a major challenge not only in Brazil [7,77] but in several developing countries [12,14,78], and this happens as a result of the accelerated urbanization process that puts pressure on urban density and segregates the poorest part of the population into informal housing in the outskirts of cities [31]. Currently, although social housing policy has been developed to provide the basic conditions for the largest possible share of this population, it often fails to provide basic infrastructure [7,17] and can result in family conflict, urban violence, depression, social costs [6] and irregular adaptation of housing for employment and income generation [32].

The urban quality criteria aim to reduce or mitigate the problems arising from this segregation of low-income families, requiring that the developments be located near city infrastructure, with regular public transportation, businesses such as drug stores, markets, schools, hospitals, etc. The importance of this criterion is evident, and this is proven by the results of the model since the required compliance with this criterion supersedes all other criteria.

Furthermore, these results demonstrate the need to rethink public policies aimed at the low-income population in these countries and to seek, for example, policies with a local focus that are more geared at inclusion, despite the spatial, social, and economic exclusion of these communities. To this end, China has good examples, which have been successful in including traditional communities in its accelerated industrialization process without harming their collectivity [79].

### 5.5. Water Management

By obtaining the third highest *t*-value in the model (2.578) and a significance of 0.01, hypothesis H4 adds criteria with a strong influence on the perceived quality of life of the residents of social projects. Out of the criteria that make up the hypothesis, criteria PWM1 and PWM2 relate to household savings for the resident by using GBTs that reduce water use by household equipment or reuse rainwater for secondary use. However, the greater loading is related to criterion PWM3—"permeable areas on the ground so that rainwater can seep and prevent flooding"—with 0.994 loading (Figure 2). Approximately 94% of

the variance of this indicator is explained by the construct, thus demonstrating that this is the criterion that explains the largest percentage of variance in the construct [50]. This is related to the high incidence of flooding in the Amazon region.

In support of this statement, Paumgarten, Maués, and Rocha [80] observed that a large part of the cities in the region is located in areas that are potentially floodable and unsuitable for housing use, areas considered by the study to have a low or medium flood risk index (due to their proximity to coastal areas and drainage channels) where the majority of this population suffers frequent (monthly) flooding and damage to furniture, equipment, cars, or difficulty of access to their homes during these events.

Indeed, according to Koerth et al. [81], the personal experience of the residents is an important trigger for the implementation of flood risk reduction measures, especially those of low effort and cost. In this regard, two hypotheses can be drawn: a) there seems to be wide acceptance of GBT related to flood risk reduction in regions with flood-prone geographical features; b) the implementation of GBT related to flood risk reduction can be a trigger for the implementation of sustainable measures in regions with flood-prone geographical features.

### 5.6. Piped Gas and Water Heating

Out of these, hypothesis H5, "piped gas and water heating" with a *t*-value of 0.286 and *p*-value of 0.765, was shown to have the least influence on residents' perceived quality of life. This can be interpreted as a cultural factor stemming from the very low occurrence of buildings with these GBTs in the region. This argument is corroborated by a fact that occurred in one of the surveyed developments, where the housing development residents' assembly decided to isolate the collective gas system to the detriment of the use of 13 Kg cylinders, broadly adopted in the region.

This can lead to two distinct understandings. The first is to discuss the obligation imposed by the SCA to have piped gas in the household since this criterion has no influence on the resident's quality of life, and this requirement leads to increased costs for the certification process without clear benefits. Second, it may represent a "socially neglected effect" of this GBT.

The term "socially neglected effect" was used by Roman, Pardo, and Irazoque [28], referring to the non-acceptance of a GBT in favor of a conventional technology already known by the user, and one of the causes of this effect is the lack of information about the benefits and functionalities thereof [25,28]. In this regard, this result suggests that the criterion should be reconsidered, and as a starting point, it is necessary to conduct further research to evaluate the need for the implementation of piped gas systems in social housing in regions with a tropical climate similar to that in northern Brazil.

### 5.7. Sustainable Leisure Equipment in the Housing Development

This was another hypothesis that was rejected (H3), with a *t*-value of 1.242 and *p*-value of 0.214, despite the presence of leisure facilities being considered one of the basic criteria for social housing [16,82] and outdoor activities having a positive influence on well-being [83].

This contradiction may be related to the very subjectivity of the concepts of well-being and quality of life, inasmuch as each individual perceives the importance of leisure equipment in the enterprise in a different manner, according to their economic situation, preferences, needs, use, and access to these spaces [84].

In fact, there is no in-depth understanding in the reference literature on the subject, and the contrast between the need to make leisure facilities available in social housing and the lack of perception about the importance of this GBT remains without an explanation. While there is evidence of dissatisfaction, insecurity, and a low percentage of use of these spaces, adopting this GBT is encouraged, and its inadequacy or inexistence often results in the adaptation of spaces such as sidewalks and garages in order to provide spaces for children to play [75,85]. In this regard, it is worth mentioning that most of the interviewees (>70%) live in housing developments where these spaces are provided, and to understand

the real nature of this phenomenon, it is suggested that research be conducted to address such subjectivity with an integrated view of engineering, architecture, and psychology.

## 6. Conclusions

The need to include the low-income population in the discussion of more sustainable social housing is undeniable due to the impact on the quality of life, health, employment, and income. Nevertheless, offering sustainable and affordable housing to low-income populations is not easy to do due to the cost and the lack of familiarity and trust of households with green building technologies [24–26]. This study aims to evaluate the influence of green building criteria on the perceived quality of life of low-income housing residents.

In sum, the results of the structural model evaluation demonstrate that there is a greater predisposition of low-income residents to perceive sustainable criteria related to more tangible aspects of sustainability, such that confer household savings (H1 and H2, with *t*-values of 2.641 and 2.067, respectively), social practices for training and generating employment and income (H7 with *t*-value of 2.291) and measures for reducing risks related to phenomena that have a high impact on their well-being, such as the risk of flooding (H4 with *t*-value of 2.578).

Urban quality, especially, was perceived as the sustainability criterion with the greatest influence on respondents' perception of quality of life (H8 with a *t*-value of 3.007). In fact, this result shows that access to city infrastructures, such as public transportation, sewage, drinking water, and health, are priority factors when conceiving and planning real estate developments for low-income populations before seeking to meet the characteristics of sustainable development, as it is a human right; therefore, it should be seen as such. For this, it is necessary to rethink public policies of access to housing for the low-income population to promote the inclusion of these families in society, first in relation to the location of the housing and only afterward in relation to social, economic, environmental, and governance aspects, for example. This will contribute to a more sustainable future for cities with social equity.

Otherwise, intangible issues such as the environmental protection criteria related to the construction of the building are not capable of influencing their perception of quality of life in the project, statistically proven by the rejection of the H6 with the *t*-value of 0.562, below 1.96, the lower limit to 5% significance [50]. The lack of importance given to environmental protection indicates the need to apply information strategies and environmental education to the population, defined by target 4.7 of the SDG as a key strategy for sustainable development. In order for low-income populations to better understand the importance of environmental protection, it is necessary not only to include environmental education in schools but also, in the case of residents of housing projects that have GBT, it is necessary to include information on how to operate these technologies in the technical manual of the project as well as provide courses on how to operate them, as understanding the importance of environmental protection and having technical, operational knowledge of these technologies is the best way to achieve efficient use of GBT so as to render the global goals set by the SDGs feasible.

Notably, the SCA has criteria that oblige the construction company to offer the GBT operation mode in the technical manual handed out to the residents, and it scores criteria related to environmental education courses and training of residents for the management of the development, with course hours of at least 4 and 12 h, respectively. These and other criteria broken down in Table 1 make up the confirmed hypothesis H7—"social practices"—perceived as very important for the quality of life in the housing development, as for widespread labels, such as BREEAM International new constructions and LEED for Homes v4 for example, they pay little attention to these criteria. The BREEAM does not even glimpse any of these issues, and the LEED, despite requiring that informative manuals be handed out to residents on existing GBTs, only requires a minimum of 1 h of training on the operation and maintenance of these technologies. Neither does it foresee any of the other criteria that make up the category of social practices of the SCA. This not only affirms the

poor adaptability of the main international labels for low-income housing but also suggests the importance of these labels in adopting criteria related to the environmental information and education of the residents as a necessity to make the current GB model compatible with the sustainable development that humanity aims to achieve, as materialized in the SDGs.

At the regional level, this lack of a keen perception of the need for environmental protection represents a barrier not only to the spreading of GBT and GB but also suppresses the potential of grassroots movements of Latin American communities for sustainability. It happens because a community that does not understand the importance of environmental protection to their household does not demand actions for the benefit of the environment from its political representatives. In order to overcome this barrier, it is necessary to strengthen the teaching of environmental education in schools, as well as to increase media exposure about the need for environmental protection for sustainable development.

Education, however, is a strategy with a long-term pay-off, so one can hardly expect to harness the mobilization potential of these communities in time to help meet the 2030 goals. This means that the only way to get these communities to align with the SDGs is through public policies, such as social housing certification, by adopting, for example, in the Brazilian case, the SCA in MCMV works.

**Author Contributions:** Conceptualization, W.B.S., methodology, W.B.S. and L.M.F.M.; software, W.B.S.; validation, W.B.S.; formal analysis, W.B.S. and L.M.F.M.; resources, W.B.S. and L.M.F.M.; writing—W.B.S.; writing—reviewand editing, W.B.S. and L.M.F.M.; visualization, W.B.S. and L.M.F.M.; supervision, L.M.F.M.; project administration, W.B.S. and L.M.F.M.; funding acquisition, L.M.F.M. All authors have read and agreed to the published version of the manuscript.

**Funding:** The author received no financial support for the research, but received financial support for the publication of this article from PROPESP/PAPQ/UFPA and financed in part by the Coor-denação de Aperfeiçoamento de Pessoal de Nível (CAPES) [Coordination for the Improvement of Higher Education Personnel]-Brazil-Finance Code 001.

**Institutional Review Board Statement:** The study was conducted in accordance with the Declaration of Helsinki, and approved by the Ethics Committee of Federal University of Pará (protocol code 49919421.4.0000.0018 approved on 9 December 2021).

**Informed Consent Statement:** Informed consent was obtained from all subjects involved in the study.

**Data Availability Statement:** Not applicable.

**Conflicts of Interest:** The authors declare no conflict of interest.

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
