# Peer review of "Environmental Protection Is Not Relevant in the Perceived Quality of Life of Low-Income Housing Residents: A PLS-SEM Approach in the Brazilian Amazon"

_sustainability, doi:10.3390/su142013171_

Round 1

Reviewer 1 Report

Authors in this study explored the influence of green building criteria on the perceived quality of life of low-income housing residents.The result of the study suggests the importance of GB criteria that confer more practical benefits to low-income residents, in addition to benefits for the environment.

I found the work is well presented. The method, results and discussion are given in details and understandable to the reader.

I see the work deserve publication in the journal.

Author Response

Thanks for yours supportives comments.

Reviewer 2 Report

The abstract should be revised to highlight the research technique used as well as clearer concluding points to the research.

What about post-occupancy assessments? Can they be included in this literature study to connect individual demands to pre-planned designs, especially in domains with environmental concerns?

Please address the font size change on page 2

More information on the strategy used to choose the sample of responders is requested. Why was the data gathering sample chosen to be the most suitable sample? Please elaborate on sampling in relation to the selected population.

Please enhance the conclusions. Too short.

The paper's English grammar to be reviewed.

Author Response

No.

Reviewers’ comments

Responses

1.        

The abstract should be revised to highlight the research technique used as well as clearer concluding points to the research.

Thanks for your supportive comment.

Abstract revised. We hope it is suitable.

2.        

What about post-occupancy assessments? Can they be included in this literature study to connect individual demands to pre-planned designs, especially in domains with environmental concerns?

Thanks for your supportive comment.

This help to improved the topic 2.2 data collection. We hope it is suitable.

3.        

Please address the font size change on page 2

Thanks for your supportive comment.

4.        

More information on the strategy used to choose the sample of responders is requested. Why was the data gathering sample chosen to be the most suitable sample? Please elaborate on sampling in relation to the selected population.

Thanks for your supportive comment.

This help to improved the topic 3.1 sample details. We hope it is suitable.

5.        

Please enhance the conclusions. Too short.

Thanks for your supportive comment.

6.        

The paper's English grammar to be reviewed

Thanks for your comment.

We made major proofreading in the text concerning language. We hope it is suitable.

7.        

Is the content succinctly described and contextualized with respect to previous and present theoretical background and empirical research (if applicable) on the topic? — “can be improved”

Thanks for your comment.

We made some improvements on text and add some citations to strengthen some arguments.

We hope it is suitable.

8.        

Are the research design, questions, hypotheses and methods clearly stated? — “can be improved”

Thanks for your comment.

To clarify this question we wrote the topic 2.4 conceptual model. We hope it is suitable.

9.        

Are the conclusions thoroughly supported by the results presented in the article or referenced in secondary literature? — “can be improved”

Thanks for your comment.

We made some improvements on text and add some citations to strengthen some arguments.

We hope it is suitable.

Reviewer 3 Report

The paper introduced a questioner-based survey that investigates the perception of low-income users concerning green building criteria.

The paper is badly written and badly organized.  The questionnaire is very sallow and not convincing. 

The Analysis method and the obtained results require deeper discussion and explanations.

I believe this paper is out of the scope of the journal, It can be considered after modification in some social studies journals.

Author Response

No.

Reviewers’ comments

Responses

1.        

The paper is badly written and badly organized.  The questionnaire is very sallow and not convincing.

Thanks for your supportive comment.

When analyzing your comment, we were surprised, mainly, because our text has a lot of similarity in structure and content with other texts published in this journal. Here are some examples:

- The Influence of Environmental and Non-Environmental Factors on Tourist Satisfaction in Halal Tourism Destinations in West Sumatra, Indonesia;

- Exploring International Faculty’s Perspectives on Their Campus Life by PLS-SEM

We tried to improve some more features in our text. We hope we have reached the most important points of your evaluation.

Regarding the questionnaire, it was submitted to the ethics committee of the university to which we belong and it was approved by the researchers of the institution's evaluation committee, under the number.........

2.        

The Analysis method and the obtained results require deeper discussion and explanations.

Thanks for your supportive comment.

Improve the description of the method, we hope it agrees with your suggestions for improvement for our text.

3.        

I believe this paper is out of the scope of the journal, It can be considered after modification in some social studies journals.

Thanks for your comment.

However, we do not agree with this suggestion of publication in another journal, as in Sustainability we found some recently published articles that bring this behavioral approach of consumers to assess issues related to sustainability. To exemplify:

-        Internal Motivations, External Contexts, and Sustainable Consumption Behavior in China—Based on the TPB-ABC Integration Model;

-        Social Benefits Evaluation of Rural Micro-Landscapes in Southeastern Coastal Towns of China—The Case of Jinjiang, Fujian.

4.        

Are all the cited references relevant to the research?

Thanks for your comment.

5.        

Are the research design, questions, hypotheses and methods clearly stated?

Thanks for your comment.

6.        

Are the arguments and discussion of findings coherent, balanced and compelling?

Thanks for your comment.

7.        

For empirical research, are the results clearly presented?

Thanks for your comment.

8.        

Is the article adequately referenced?

Thanks for your comment.

Round 2

Reviewer 2 Report

The paragraph 2.5 requires a re-check of grammar.

Words use hyphens unnecessarily. Please check accordingly. 

Conclusion paragraph to be revised (first paragraph). The use of the words 'a better study' is not a clear measurement of improvement. Kindly rephrase. 

Overall concluding paragraph can be improved to explain recommendations being put forward in a measurable way. 

Author Response

No.

Reviewers’ comments

Responses

Reviewer 2

1.        

The paragraph 2.5 requires a re-check of grammar.

Thanks for your supportive comment.

We hope it is suitable.

2.        

Words use hyphens unnecessarily. Please check accordingly.

Thanks for your supportive comment.

We hope it is suitable.

3.        

Conclusion paragraph to be revised (first paragraph). The use of the words 'a better study' is not a clear measurement of improvement. Kindly rephrase. 

Thanks for your supportive comment.

We changed the content of the conclusion, making it clearer as to the fulfillment of the research objective, also quantifying the results.

We hope it is suitable.

4.        

Overall concluding paragraph can be improved to explain recommendations being put forward in a measurable way. 

5.        

Are the research design, questions, hypotheses and methods clearly stated? — “can be improved”

Thanks for your comment.

We made modifications that we deemed relevant, mainly highlighting the research objectives. Furthermore, we understand that design research is adequate. We hope it is suitable.

Reviewer 3 Report

I still have my concerns regarding the structure of the paper. For example, the introduction has 4 subsections and the conclusion is 2 pages which is very clearly not the best structure.

The English written has been enhanced, however, it still needs more improvements

I insist on regarding my last comment about considering this paper in a social science journal. I cannot see any technical or scientific contribution in this paper makes suitable for Journal.

Author Response

No.

Reviewers’ comments

Responses

Reviewer 3

1.        

I still have my concerns regarding the structure of the paper. For example, the introduction has 4 subsections and the conclusion is 2 pages which is very clearly not the best structure.

Thanks for your comment.

We have remodeled the structure of the paper to meet your requirement, we believe that the article is now much better.

For this we created the literature review topic 2 and made changes to the structure of the conclusion. We hope it now meets your requirements.

2.        

The English written has been enhanced, however, it still needs more improvements

Thanks for your comment.

We did a new English revision. The professional responsible was the same one who translated other papers that we published in Sustainability magazine.

3.        

I insist on regarding my last comment about considering this paper in a social science journal. I cannot see any technical or scientific contribution in this paper makes suitable for Journal.

Thanks for your comment.

We respect your observation in this regard, but to reinforce the argument that this paper is adherent to being published in Sutainability, we cite more similar works:

-        Environmental Sustainability Commitment and Access to Finance by Small and Medium Enterprises: The Role of Financial Performance and Corporate Governance

-        What Drives Social Enterprises to Form Sustainable Values? The Effects of Normative Identity and Social Performance

-        Do Governance Perceptions Affect Cooperativeness? Evidence from Small-Scale Irrigation Schemes in Northern Ghana

-        From ESG to DESG: The Impact of DESG (Digital Environmental, Social, and Governance) on Customer Attitudes and Brand Equity

-        Evaluating Perception of Sustainability Initiatives Invested in the Coastal Area of Versilia, Italy

4.        

Is the content succinctly described and contextualized with respect to previous and present theoretical background and empirical research (if applicable) on the topic? — “must be improved”

Thanks for your comment.

5.        

Are the research design, questions, hypotheses and methods clearly stated? — “can be improved”

Thanks for your comment.

6.        

Are the arguments and discussion of findings coherent, balanced and compelling? — “must be improved”

Thanks for your comment.

7.        

Is the article adequately referenced? — “must be improved”

Thanks for your comment.

8.        

Are the conclusions thoroughly supported by the results presented in the article or referenced in secondary literature? — “must be improved”

Thanks for your comment.

Round 3

Reviewer 3 Report

1- The paper still needs more improvements regarding the structure of the paper, mainly the literature review and the conclusion sections, Figure 1 should be re-painted in better resolution and its caption should be connected to the figure, not on a separate page, avoid having two titles of a section and a subsection directly.

2- The privileges and advantages of each question in the survey should be explicitly explained.

3- The obtained benefits and find results of the survey should be graphically represented in the last section.

4- Maybe the paper topic is related to the journal since it has some previously published papers with the same objective. But the current version of the paper needs many enhancements to be accepted in this journal.

Author Response

No.

Reviewers’ comments

Responses

1.        

1- The paper still needs more improvements regarding the structure of the paper, mainly the literature review and the conclusion sections avoid having two titles of a section and a subsection directly.

Thanks for your comment.

We include a short introduction in chapters 3 and 4.

About the chapters “literature review” and “conclusion” we do not understand what exactly we could improve. For now, we understand that the improvements requested in lasts rounds improved satisfactory. We hope its suitable.

2.        

1.1 - Figure 1 should be re-painted in better resolution and its caption should be connected to the figure, not on a separate page

Thanks for your comment.

We re-paint the figure 1 in 1000x1000 resolution, according the “information for authors” of the journal.

Besides, we improve the structure of the text to better connect to figure 1, approximating it with it citation.

3.        

2- The privileges and advantages of each question in the survey should be explicitly explained.

Thanks for your comment.

We include the column “benefits” in table 1, addressing the advantages of each question.

Its important to note that the advantages included was based on Blue House Label manual.

4.        

3- The obtained benefits and find results of the survey should be graphically represented in the last section.

Thanks for your comment.

We included figure 2, containing a graphical representation of PLS-SEM results. According to others papers of the journal, for example:

§  Product Innovation, Market Intelligence and Pricing Capability as a Competitive Advantage in the International Performance of Startups: Case of Peru

§  Absorption Capacity and Development of Photocatalyst Green Ceramic Products with Moderation of Green Environment for Sustainability Performance of Developing Industries

§  Exploring the Relationship between Data Analytics Capability and Competitive Advantage: The Mediating Roles of Supply Chain Resilience and Organization Flexibility

5.        

4- Maybe the paper topic is related to the journal since it has some previously published papers with the same objective. But the current version of the paper needs many enhancements to be accepted in this journal.

Thanks for your comment.

Yours reviews helps a lot. As well as the others two reviews, witch approve the paper in last rounds.

We agree that the article now has a better structure, which makes it easier to understand. As well as that the method and objectives of the research are clearer.

We believe that the article now meets the standards of the journal, and knowing the importance of discussing the sustainability of housing for the low-income population, we hope that the article will be published. Thanks.

Round 4

Reviewer 3 Report

The paper has been improved.

The quality of figures and results representations still needs more work.

Author Response

No.

Reviewers’ comments

Responses

1.        

The quality of figures and results representations still needs more work.

Thanks for your comment.

We improved the figure 2 to better represent the results and avoid misunderstanding.

About the resolution, actually the 2 figures are in significant better resolution than the minimum recommendation for author of the journal:

Actual figures resolution: 8358x4150 pixels and 1000 dpi

Paper minimum recommendation: 1000x1000 pixels and 300dpi

Its possible that Microsoft Word reduce the quality of the figures, in this regards, we attach the two figures in full resolution. The editor authorized this proceeding. That way the journal team will deal directly with the figures files.
